# Missed Opportunities for HIV Diagnosis and Their Clinical Repercussions in the Portuguese Population—A Cohort Study

**DOI:** 10.3390/pathogens13080683

**Published:** 2024-08-13

**Authors:** João Lourinho, Maria João Miguel, Frederico Gonçalves, Francisco Vale, Cláudia Silva Franco, Nuno Marques

**Affiliations:** 1Infectious Diseases Department, Hospital Garcia de Orta, ULS Almada-Seixal, 2805-267 Almada, Portugal; maria.joao.miguel@ulsas.min-saude.pt (M.J.M.); joao.goncalves@ulsas.min-saude.pt (F.G.); francisco.vale@ulsas.min-saude.pt (F.V.); claudia.franco@ulsas.min-saude.pt (C.S.F.); 2Clínica de Doenças Infecciosas, Faculdade de Medicina, Universidade de Lisboa, Avenida Professor Egas Moniz, 1649-028 Lisboa, Portugal; nmdsmarques@gmail.com; 3ULS da Arrábida, 2910-549 Setúbal, Portugal

**Keywords:** HIV, missed opportunities, AIDS-defining conditions, HIV transmission

## Abstract

Late human immunodeficiency virus (HIV) diagnosis has been associated with missed opportunities for earlier diagnosis. We conducted a retrospective, longitudinal, single-centre cohort study evaluating these missed opportunities and their clinical repercussions in adults with a new HIV diagnosis or who were drug-naïve, who attended our Infectious Diseases Department between 2018 and 2023. We assessed missed opportunities in the two years prior to diagnosis or after the last negative HIV test. We compared clinical and laboratorial data from individuals with and without missed opportunities. The primary outcome considered was AIDS-defining conditions at diagnosis. Among the 436 included individuals, 27.1% experienced at least one missed opportunity. Those with missed opportunities were more likely to be female (*p* = 0.007), older at their first consultation (*p* < 0.001), born in Africa (*p* < 0.001) and in countries with a high HIV prevalence (*p* < 0.001), and have heterosexual transmission (*p* < 0.001). The adjusted analysis showed that missed opportunities were significantly associated with AIDS-defining conditions at diagnosis (OR 3.23, CI 95% [1.62–6.46], *p* < 0.001). These findings highlight the impact of missed opportunities on HIV severity, underscoring the need for more targeted interventions to reduce them.

## 1. Introduction

The ongoing human immunodeficiency virus (HIV) epidemic continues to affect the health of many individuals in Europe, with millions of newly diagnosed and reported cases documented over recent decades [1]. Despite new diagnoses and the favourable trend indicating decreasing numbers of people living with undiagnosed HIV infection over time, there is still a significant number of such cases in Europe, 33% of which have advanced HIV infection [2].

Late HIV diagnosis is a serious issue in high-resource countries, leading to considerable morbidity and mortality burdens which could potentially be avoided by earlier diagnosis and treatment [3]. Factors that are traditionally associated with late presentation include non-men who have sex with men, older age, and non-European origin [4]. 

Late diagnosis has been associated with pre-diagnosis encounters with healthcare providers where an HIV test was indicated but not ordered, thereby resulting in missed opportunities for earlier HIV detection. There is a growing body of evidence indicating that individuals in high-resource settings continue to be susceptible to missed opportunities to HIV diagnosis [5,6,7,8,9,10,11,12,13]. 

The Portuguese guidelines for the diagnosis of HIV infection recommend an HIV serology in all adults up to 64 years old in an opportunistic manner. Furthermore, the Portuguese guidelines provide for testing according to various clinical and epidemiological criteria (Table 1) [14].

Despite HIV testing being free of charge in Portugal, in 2022, 57.2% of new diagnoses of HIV infection had a CD4 count ≤ 350 cells/mm^3^, revealing a late presentation to clinical care, and 34.2% of new diagnoses had ≤ 200 cells/mm^3^, an indicator of advanced disease [15].

This strongly indicates existing opportunities to optimize the Portuguese HIV testing strategy. Understanding the missed opportunities for diagnosing HIV infection seems fundamental to optimize this strategy and to identify people living with HIV (PLWH) earlier in the course of the infection, minimizing the adverse outcomes related to late diagnosis of the disease.

We propose a study to identify missed opportunities for HIV diagnosis and the population that is most susceptible to such oversights and to evaluate the clinical implications upon patient presentation at the time of diagnosis.

## 2. Materials and Methods

### 2.1. Design, Setting, Participants, and Definitions

We conducted a retrospective, longitudinal, single-centre cohort study evaluating missed opportunities for HIV diagnosis and their clinical repercussions. 

We defined missed opportunities for HIV diagnosis as any pre-diagnosis encounter with healthcare, where an HIV test was indicated but was not ordered. Indication for HIV testing was established according to the Portuguese guidelines from 2014, the Direção-Geral da Saúde guideline for HIV diagnosis and laboratory screening (Table 1). We assessed the presence of missed opportunities in the two years prior to diagnosis, or after the last negative HIV test if it had occurred less than two years prior to diagnosis.

Whenever two different indications for HIV testing were present in the same consultation, we considered them to be two separate missed opportunities.

We defined high-prevalence countries as countries with an HIV prevalence higher than 1% [16].

Epidemiological criteria were not considered missed opportunities when detected in the emergency department, considering the difficulties experienced in this setting, namely the limited time available for individual observation and the need to allocate complementary diagnostic resources for urgent situations. An exception was applied to victims of sexual violence, due to its urgent character.

Potentially eligible patients included adult individuals with a new HIV diagnosis or previously diagnosed individuals who were drug-naïve, meaning that they had never taken any antiretroviral therapy before. Individuals considered were patients who had a first consultation in our Department of Infectious Diseases, in a tertiary hospital in Almada, Portugal. Enrolled patients were selected during a six-year timespan, from January 2018 to December 2023. We used a consecutive sampling method. The total number of individuals meeting our criteria determined the sample size. Individuals with a first consultation before 2018 or who had already undergone antiretroviral therapy before were excluded.

We compared clinical and laboratorial data from individuals with missed opportunities for HIV diagnosis (study group) with those without missed opportunities (control group). 

We followed the Strengthening the Reporting of Observational studies in Epidemiology (STROBE) guidelines for reporting.

### 2.2. Outcomes and Variables

The primary outcome considered was acquired immunodeficiency syndrome (AIDS)-defining conditions at the time of HIV diagnosis. As a secondary outcome, the CD4 count at the time of diagnosis was compared between both groups, specifically a CD4 count ≤ 200 cells/mm^3^ and a CD4 count ≤ 350 cells/mm^3^.

Relevant demographic, laboratory, and clinical data were obtained by consulting individual digital files. The considered demographic variables were sex; date of birth; nationality; duration of stay in Portugal; principal risk factor for HIV infection; registration in a primary healthcare centre; and assignment of a general practitioner. The considered laboratory variables were date of the last negative HIV serology; type and subtype of HIV; CD4 count and HIV viral load at diagnosis; serologies for hepatitis B virus (HBV) and hepatitis C virus (HCV); and total white blood cell, lymphocyte, and platelet counts in the two years prior to diagnosis. The considered clinical variables were date of HIV infection diagnosis; date of the first consultation in our department; existence of opportunistic infections at the time of HIV diagnosis or during the two years prior to diagnosis; number and reason for visits to hospital emergency department in the two years prior to the diagnosis date; and number and reason for visits to the primary healthcare centre in the two years prior to the diagnosis date. 

In the case of patients where access to the primary healthcare centre data was unattainable, we considered them as not having missed opportunities to HIV diagnosis in the primary healthcare setting. The reasons for not being able to access data were (1) death prior to the study commencement, (2) a lack of authorization for access to those data, and (3) an absence of a valid health user number or passport. We did not exclude these patients to reduce selection bias. We considered that they did not have missed opportunities in the primary healthcare setting because, by definition, missed opportunities involve encounters with healthcare, which we could not prove occurred in these cases. It is noteworthy that these patients may have had missed opportunities in the hospital setting.

### 2.3. Statistical Analysis

Categorical variables were reported using frequencies and percentages and continuous variables using means and standard deviation (SD) or median and interquartile range (IQR), when skewly distributed. Normality was tested using Kolmogorov–Smirnov and Shapiro–Wilk tests. Comparison across groups was performed using Chi-square tests for categorical data (Fishers’ when applicable) and *t*-tests for continuous data (Mann–Whitney U when non-normally distributed). Adjusted analysis was conducted using binary multivariate logistic regression, and the independent variables included in the model were previously chosen based on biological and clinical plausibility. Odds ratio (OR) and 95% confidence intervals (CIs) for primary outcome were reported. Statistical analysis was carried out using SPSS 29.0.0.0 for iOS (IBM SPSS Statistics). We considered a *p* value < 0.05 statistically significant.

## 3. Results

### 3.1. Study Population

Among the 1115 individuals referred for an HIV Infectious Diseases consultation in our department between January 2018 and December 2023, we included 436 in our study, while 679 were excluded because they had already started or undergone antiretroviral therapy previously (Figure 1). From our sample, 27.1% (118/436) of individuals experienced at least one missed opportunity for HIV diagnosis and were included in the study group. The remaining 72.9% (318/436) were included in the control group.

The relevant demographic, social, and clinical data of the total sample and both study and control groups are detailed in Table 2. Several statistically significant differences were found between the two groups. Being female was found to be more prevalent in the study group (45.8%, 54/118) than in the control group (31.8%, 101/318) (*p* = 0.007). The age at the first consultation was found to be higher in the study group (42 years, IQR 33–55) than in the control group (36 years, IQR 29–45.3) (*p* < 0.001). Being born in an African country was more prevalent in the study group (44.1%, 52/118) than in the control group (27.4%, 87/318) (*p* < 0.001), and being born in a high-prevalence country occurred more frequently in the study group (36.4%, 43/118) than in the control group (15.1%, 48/318) (*p* < 0.001). Heterosexual transmission was also found to be more frequent in the study group (69.9%, 72/118) than in the control group (50.9%, 148/318) (*p* < 0.001). It was also found that the study group had a higher occurrence of having an assigned general practitioner (62.9%, 73/118) compared with the control group (46.2%, 134/318) (*p* = 0.002) and a higher number of visits to the hospital emergency department (1, IQR 0–3) than in the control group (0, IQR 0–1) (*p* < 0.001).

### 3.2. Missed Opportunities

A total of 192 missed opportunities were found in our study (Table 3). The most frequent missed opportunities included being born in a country with an HIV prevalence >1% (20.8%, 40/192), mononucleosis-like syndromes (17.2%, 33/192), and other sexually transmitted diseases (15.6%, 30/192). In our study, 63.5% of missed opportunities occurred in primary healthcare settings.

### 3.3. AIDS-Defining Conditions

A total of 104 AIDS-defining conditions at the time of HIV diagnosis were found in 17.4% (76/436) of the included individuals (Table 4).

### 3.4. Primary and Secondary Outcomes

AIDS-defining conditions at the time of HIV diagnosis occurred in 29.7% (35/118) of individuals with missed opportunities and in 12.9% (41/318) of individuals without missed opportunities for HIV diagnosis (*p* < 0.001) (Table 5). After adjustment for confounding factors, missed opportunities predicted the occurrence of AIDS-defining conditions (OR 3.23, CI 95% [1.62–6.46], *p* < 0.001). The potential confounding factors considered were age, sex, nationality, heterosexual transmission, type of HIV, assignment of a general practitioner, and number of visits to hospital emergency department.

As for our secondary outcomes, we found that a CD4 count ≤ 200 cells/mm^3^ at the time of diagnosis occurred in 41.5% (49/118) of the individuals in the study group and 30.3% (96/318) in the control group (*p* = 0.027). We found no statistically significant differences in the occurrence of a CD4 count ≤ 350 cells/mm^3^ between the two groups (Table 5). 

## 4. Discussion

Our single-centre study shows that missed opportunities for diagnosing HIV infection were associated with a significantly higher risk of the presence of AIDS-defining conditions at HIV diagnosis, which was maintained after adjusting for confounding factors (OR 3.23, CI 95% [1.62–6.46], *p* < 0.001). These results suggest that individuals who suffer from missed opportunities have an increased risk of developing AIDS-defining conditions at diagnosis. This seems to reflect that missed opportunities lead to a delay in diagnosis and to a more severe immunosuppression. A Canadian study showed a significant association between missed opportunities and AIDS diagnosis within three months of HIV diagnosis but did not distinguish between CD4 counts below 200 cells/mm^3^ and AIDS-defining conditions [17]. A Scottish study also demonstrated a significant association between AIDS-defining conditions and patients with missed opportunities [8]. Neither of these studies adjusted their results for possible confounding factors. To our knowledge, this is the first study to do so. A German study demonstrated no significant association between missed opportunities and AIDS-defining conditions. However, the study population was only late presenters, which may overestimate the number of AIDS-defining conditions in their sample [9]. Since highly active antiretroviral therapy was introduced, mortality in PLWH has decreased substantially. However, death from AIDS still occurs in individuals with late diagnosis and in those who have difficulty engaging in care or adhering to therapy [3]. An English study reported that AIDS continues to account for the highest proportion of deaths despite the availability of free HIV treatment and care [18]. This highlights the importance of early HIV diagnosis to maximize the individual and public health benefits of antiretroviral treatment. Our data demonstrate the clinical relevance of missed opportunities and their impact on the clinical severity of HIV, underscoring the need to make efforts to reduce their existence.

Regarding AIDS-defining conditions, *Pneumocystis jirovecii* pneumonia stands out as the most common, followed by tuberculosis, wasting syndrome due to HIV, and oesophageal candidiasis. These findings align with most recent national and European data, which reported *Pneumocystis jirovecii* pneumonia as the most frequent AIDS-defining condition, accounting for 26.8% of cases in Portugal and 21.2% in Europe [1,15]. As for the remaining AIDS-defining conditions, although both sets of data report similar conditions, the prevalence varies. Similarly to national data, we observed tuberculosis to be the second most prevalent AIDS-defining condition. European data, however, report HIV wasting syndrome as the second most prevalent condition. We also observed a high proportion of HIV wasting syndrome in our population, even though it accounts for only 5.8% of patients in Portuguese data [1,15]. These differences underline the need for tailored healthcare strategies to address the specific epidemiological patterns in each region. 

We also showed that missed opportunities are significantly associated with a CD4 count of ≤ 200 cells/mm^3^ at diagnosis (*p* = 0.027). However, we did not find a significant association between missed opportunities and a CD4 count of ≤ 350 cells/mm^3^ at diagnosis (*p* = 0.860). Previous studies have correlated missed opportunities with a lower CD4 count, but comparisons are challenging due to inconsistent use of CD4 thresholds across different studies [5,6,8,17]. Our data seem to imply that missed opportunities influence the progression to a CD4 count ≤ 200 cells/mm^3^ but not to a CD4 count ≤ 350 cells/mm^3^. We do not fully understand the meaning of these findings, but in our setting, PLWH seem to present themselves late to clinical care, even without missed opportunities. As mentioned earlier, a majority of new diagnoses of HIV in Portugal had a CD4 count ≤ 350 cells/mm^3^, which reveals a tendency for late presentation across all the Portuguese population and not only in our sample [15]. This trend may weaken the association in our study. Another possible explanation is the fact that we only looked for missed opportunities in the two years prior to diagnosis.

Our study also aligns with previous research that tries to characterize individuals who are at risk of experiencing missed opportunities. As in our study, several others have also shown that older age and heterosexual transmission are independent risk factors for experiencing missed opportunities and for late HIV diagnosis [4,6,7,8,9,12]. An Israeli retrospective cohort study identified old age as a significant risk factor for late HIV diagnosis [12]. Regarding gender, our study found a significant association between missed opportunities and being female, whereas most studies did not specifically highlight gender as a significant factor in missed opportunities for HIV diagnosis [5,6,8,9,11,12]. One study noted males predominantly [7].

Regarding geographic factors, our study noted a higher prevalence among individuals born in Africa and in high-prevalence countries. It is noteworthy that all our sample’s high-prevalence countries were in Africa, meaning that there is probably an overlap in these two associations. It is important to state that our region has one of the highest rates of migrant populations in the country. Consequently, it is not surprising that individuals from high-prevalence countries constitute a large portion of the total missed opportunities in our sample. It is also worth mentioning that being born in countries with more than 1% of HIV prevalence was by itself an epidemiological criterion on which HIV serology should be performed, which may overrepresent individuals from these countries in the study group. Other studies also describe people born in high-prevalence countries as late presenters or having missed opportunities [4,6,7,9]. While a German study observed that patients from high-prevalence countries were more likely to be late presenters [9], a European study (COHERE) found that late presentation was more likely among individuals from Africa and Southern Europe [4]. A British Columbia study similarly identified that certain demographic subgroups, such as those from remote regions (which might include immigrants from high-prevalence areas), faced higher risks of missed diagnoses [6]. Some studies, however, did not specifically highlight the country of birth as a significant factor [5,8,10,11,19].

Our study associated missed opportunities with having an assigned general practitioner and more emergency department visits. We also observed that 63.5% of missed opportunities occurred in primary healthcare settings. These findings highlight that missed opportunities occurred in all healthcare facilities, even in individuals with an assigned general practitioner who had consultations more often. A retrospective cohort study from Israel emphasized missed opportunities in both primary and secondary care settings, noting that 60% of clinical indicator diseases were missed by general practitioners and 40% by hospitalists [12]. A Scottish study emphasized missed opportunities in primary care and non-specialist settings [8]. A Moroccan study pointed out that 61% of patients had consulted a general practitioner at least once in the three years prior to diagnosis, with 88% having missed opportunities despite these consultations [19]. Several other studies also found a significant association between missed opportunities for HIV testing and frequent visits to the emergency department [5,7,11]. We emphasize, however, that missed opportunities imply an encounter with healthcare, and it was those encounters that served as the object for information collection for our study. All these findings indicate that certain groups are overlooked for HIV diagnosis, highlighting the need for increased vigilance and serological testing in these populations.

The high prevalence of missed opportunities involving mononucleosis-like syndromes, other sexually transmitted diseases, haematological abnormalities, unexplained weight loss, and bacterial pneumonia may reflect a lack of awareness among the medical community regarding differential diagnoses and the criteria for requesting HIV serology in our region. Notably, several of these most common missed opportunities are part of the indicator conditions by which HIV testing should be guided according to the HIDES studies [20,21]. Therefore, our study highlights the most common missed opportunities and can serve as a foundation for community intervention.

Unaware HIV-positive individuals have higher transmission rates, which is a public health concern [22]. Also, missed opportunities are associated with increased direct healthcare costs [17]. In our sample, 27.1% of individuals experienced at least one missed opportunity. Previous studies have demonstrated an incidence of missed opportunities between 14 and 26% [6,8,11,17]. Other studies, whose population were late presenters, have higher percentages, between 21 and 77%, but usually have smaller sample sizes, and the results are more difficult to generalize [5,9,10]. However, differences in the definition of missed opportunities used in each study make them difficult to compare directly. We believe that establishing a universal definition of missed opportunities would aid in the goal of eradicating HIV infection by effectively targeting neglected population subgroups, thus breaking the transmission chain earlier, with positive outcomes in health and economics.

Regarding future prospects, we propose the implementation of measures to reduce the number of missed opportunities. In that regard, it would be useful if other centres studied their own missed opportunities to better characterize them globally. A thorough characterization of missed opportunities, the population subgroups, and the places where they are more prevalent can serve as a foundation for more targeted interventions in the community.

There are several limitations to report in the present study. The retrospective nature led to several gaps in the information collected. This is exacerbated by the fact that we were unable to access all of the individuals’ medical visits, particularly consultations in private healthcare facilities. This may underestimate the number of individuals with missed opportunities. Furthermore, we chose to assess missed opportunities in the two years prior to diagnosis; however, given the number of late presenters observed in our sample, an assessment over a longer period could have been more pertinent for our sample. As previously mentioned, we considered two different missed opportunities whenever two different indications for HIV testing were present in the same consultation. We chose this approach to highlight the fact that an HIV serology test was recommended to be requested individually for each condition. We know that this overestimates the number of missed opportunities, but it does not overestimate the number of patients with missed opportunities. Finally, we did not evaluate long-term follow-up data, which could be important to clarify the long-term severity of missed opportunities, addressing other relevant outcomes such as morbidity and mortality.

## 5. Conclusions

We conclude that individuals with missed opportunities for HIV diagnosis have higher odds of developing an AIDS-defining condition at HIV diagnosis, which reflects the clinical severity of the missed opportunities. A thorough characterization of missed opportunities and the affected population subgroups can help address the HIV epidemic by serving as a foundation for more targeted interventions, both in the medical community and in the non-medical community.

## Figures and Tables

**Figure 1 pathogens-13-00683-f001:**
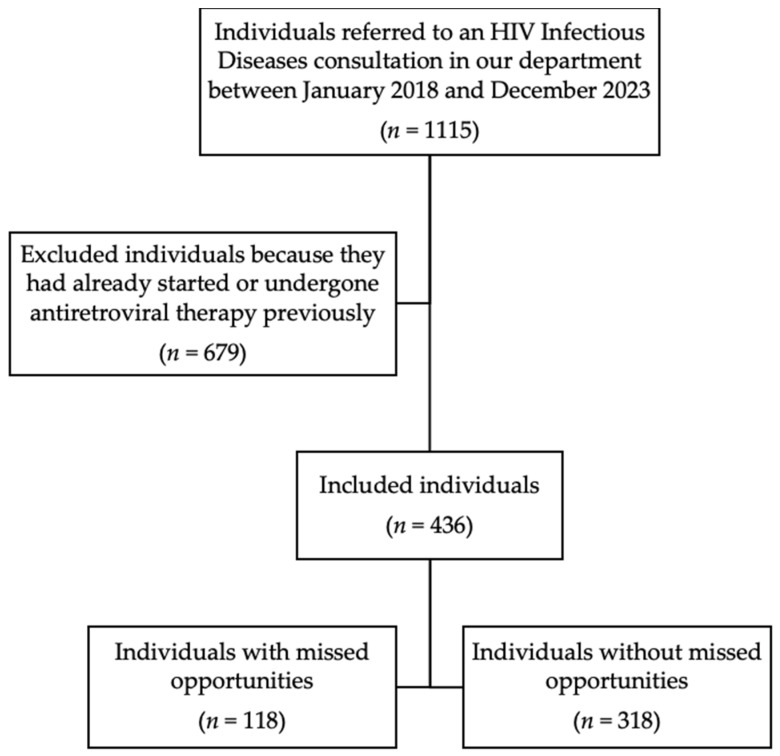
Flowchart of inclusion and exclusion process.

**Table 1 pathogens-13-00683-t001:** Clinical and epidemiological criteria on which HIV serology should be performed, according to Portuguese guidelines [14].

Epidemiological criteria	All adults up to 64 years old in opportunistic manner	
	Sexual partners of individuals diagnosed with HIV infection	
	Men who have sex with men	
	Sexual partners of men who have sex with men	
	Individuals with history of drug use	
	Individuals originating from countries with high prevalence of HIV infection (> 1%)	
	Sexual partners of individuals originating from countries with high prevalence of HIV infection	
	Inmates	
	Nomad population	
	Homeless individuals	
	Sex workers	
	Individuals subjected to sexual violence	
Clinical criteria	Any AIDS-defining condition	
	Pneumological	Bacterial pneumonia
		Aspergillosis
	Neurological	Aseptic meningitis/encephalitis
		Cerebral abscess
		Space-occupying lesion of unknown cause
		Guillain–Barré syndrome
		Transverse myelitis
		Peripheral neuropathy
		Dementia
		Leukoencephalopathy
	Dermatological	Severe seborrheic dermatitis
		Severe psoriasis
		Recurrent or multidermatomal herpes zoster
	Gastroenterological	Oral candidiasis
		Chronic diarrhoea of unknown cause
		Weight loss of unknown cause
		Infection with *Salmonella* spp., *Shigella* spp. or *Campylobacter* spp.
		Hepatitis B or C virus infection
	Oncological	Anal intraepithelial neoplasia or dysplasia
		Lung neoplasm
		Seminoma
		Head and neck neoplasm
		Hodgkin’s lymphoma
		Castleman’s disease
	Gynaecological	Vaginal intraepithelial neoplasia
		Grade 2 or higher cervical intraepithelial neoplasia
	Haematological	Thrombocytopenia, leukopenia, lymphopenia
	Ophthalmological	Retinal diseases, including herpes viruses or Toxoplasma
		Any unexplained retinopathy
	Ears, nose and throat	Lymphadenopathy of unknown cause
		Chronic parotitis
		Lymphoepithelial parotid cysts
	Others	Mononucleosis-like syndromes
		Fever of unknown cause
		Any lymphadenopathy of unknown cause
		Any sexually transmitted infection

**Table 2 pathogens-13-00683-t002:** Population’s baseline characteristics.

Characteristic	Total (*n* = 436)	Without Missed Opportunities (*n* = 318)	With Missed Opportunities (*n* = 118)	*p* Value
Female—*n* (%)	155 (35.6)	101 (31.8)	54 (45.8)	0.007
Age, years—median (IQR)	38 (29–47)	36 (29–45.3)	42 (33–55)	<0.001
Birthplace, by continent—*n* (%)				<0.001
Europe	199 (45.6)	145 (45.6)	54 (45.8)	
Africa	139 (31.9)	87 (27.4)	52 (44.1)	
America	97 (22.2)	85 (26.7)	12 (10.2)	
Asia	1 (0.2)	1 (0.3)	0 (0.0)	
High-prevalence countries—*n* (%)	91 (20.9)	48 (15.1)	43 (36.4)	<0.001
Heterosexual transmission—*n* (%)	220 (55.8)	148 (50.9)	72 (69.9)	<0.001
Assigned general practitioner—*n* (%)	207 (51.0)	134 (46.2)	73 (62.9)	0.002
Visits to the emergency department—median (IQR)	0 (0–1)	0 (0–1)	1 (0–3)	<0.001
HIV type 1—*n* (%)	419 (96.1)	308 (96.9)	111 (94.1)	0.182
HIV viral load at diagnosis, log—median (IQR)	4.8 (4.0–5.5)	4.8 (4.0–5.4)	5.0 (4.0–5.6)	0.255
HBV coinfection—*n* (%)	19 (4.4)	13 (4.1)	6 (5.1)	0.652
HCV coinfection—*n* (%)	11 (2.6)	10 (3.2)	1 (0.9)	0.302

**Table 3 pathogens-13-00683-t003:** Missed opportunities for HIV diagnosis.

Missed Opportunities for HIV Diagnosis			*n* (%)
Epidemiological criteria	Individuals originating from countries with high prevalence of HIV infection (>1%)		40 (20.8)
	Individuals with history of drug use		4 (2.1)
	Men who have sex with men		1 (0.5)
	Individuals subjected to sexual violence		1 (0.5)
	Sexual partners of individuals diagnosed with HIV infection		1 (0.5)
Clinical criteria	Pneumological	Bacterial pneumonia	14 (7.3)
	Neurological	Aseptic meningitis/encephalitis	1 (0.5)
	Dermatological	Recurrent or multidermatomal herpes zoster	6 (3.1)
		Severe psoriasis	1 (0.5)
	Gastroenterological	Weight loss of unknown cause	17 (8.9)
		Oral candidiasis	3 (1.6)
		Chronic diarrhoea of unknown cause	3 (1.6)
	Gynaecological	Grade 2 or higher cervical intraepithelial neoplasia	1 (0.5)
	Haematological	Thrombocytopenia, leukopenia, lymphopenia	27 (14.1)
	Others	Mononucleosis-like syndromes	33 (17.2)
		Fever of unknown cause	1 (0.5)
		Any lymphadenopathy of unknown cause	8 (4.2)
		Any sexually transmitted infection	30 (15.6)

**Table 4 pathogens-13-00683-t004:** AIDS-defining conditions.

AIDS-Defining Conditions	*n* (%)
Candidiasis of the oesophagus	13 (12.5)
Cryptococcosis, extrapulmonary	2 (1.9)
Cryptosporidiosis, chronic intestinal (greater than one month’s duration)	1 (1.0)
Cytomegalovirus disease (other than liver, spleen, or nodes)	7 (6.7)
Encephalopathy, HIV-related	6 (5.8)
Kaposi sarcoma	5 (4.8)
Lymphoma, immunoblastic (or equivalent term)	1 (1.0)
*Mycobacterium tuberculosis*, any site (pulmonary or extrapulmonary)	19 (18.3)
*Mycobacterium*, other species or unidentified species, disseminated or extrapulmonary	2 (1.9)
*Pneumocystis jirovecii* pneumonia	22 (21.2)
Progressive multifocal leukoencephalopathy	1 (1.0)
Toxoplasmosis of brain	10 (9.6)
Wasting syndrome due to HIV	15 (14.4)

**Table 5 pathogens-13-00683-t005:** Primary and secondary outcomes.

Outcome	Total (*n* = 436)	Without Missed Opportunities (*n* = 318)	With Missed Opportunities (*n* = 118)	*p* Value
AIDS-defining conditions at diagnosis—*n* (%)	76 (17.4)	41 (12.9)	35 (29.7)	<0.001
CD4 count ≤ 200 cells/mm^3^ at diagnosis—*n* (%)	145 (33.3)	96 (30.3)	49 (41.5)	0.027
CD4 count ≤ 350 cells/mm^3^ at diagnosis—*n* (%)	244 (56.0)	177 (55.8)	67 (56.8)	0.860

## Data Availability

The data that support the findings of this study are available on request from the corresponding author.

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
