# Peer review of "Missed Opportunities for HIV Diagnosis and Their Clinical Repercussions in the Portuguese Population—A Cohort Study"

_pathogens, 2024, doi:10.3390/pathogens13080683_

Round 1
Reviewer 1 Report
Comments and Suggestions for Authors
This manuscript investigated the missed opportunity of HIV testing in some Portuguese, and the HIV situation after diagnosis. The authors found that the population who missed opportunity had lower CD4 counts (more had CD4 counts <200 cells/μL) and AIDS-defining conditions, and more were female and were born in Africa (with high HIV prevalence). Early ART is needed for HIV-infected persons and is important for HIV prevention, but it needs early HIV testing and diagnosis. Investigating the missing HIV testing and related factors is an important issue. The findings of this study are meaningful for Portugal, even for other countries. However, I have some concerns about the analysis and discussion.
1. Method 2.2 Outcomes and variables: the reasons or reference for choosing the primary outcome and secondary outcome is needed.
2. Results 3.4: including CD4 < 200 cells/mm3 as confounding factor for the association of missed opportunity and AIDS-defining conditions seems not appropriate because CD4< 200 maybe an indicator of AIDS condition. Besides, too many factors seemed included int the multivariable analysis. There were only 76 cases (events) with AIDS-defining conditions, but the authors included 9 factors (missed opportunity which was the main factor, and age, sex, nationality, heterosexual transmission, CD4 <200, type of HIV, assignment of general practitioner, number of visits to hospital emergency department).
3. Discussion lines 235-243, the authors did not mention their results here but only discussed others’ discovery. Especially the last sentence, reference #6 did not find females were not associated with missed opportunity on multivariable analysis. However, it seems that the authors did not perform multivariable analysis for factors of the missed opportunity in this study.
4. Discussion lines 274-277,… meaning that these associations may be biased. I do not understand the “biased” mean here. So, maybe the authors can delete this part and connect directly to the next paragraph. Then it will be much easier to be read.
Minor issue
Line 172, 13.2% (42/318) should be 12.9% (41/318)?
Reviewer 2 Report
Comments and Suggestions for Authors
The authors identified significant missed opportunities for earlier diagnosis, particularly among certain demographics by analyzing data from 436 adults diagnosed with HIV between 2018 and 2023. Their findings emphasized the importance of timely testing and intervention to improve health outcomes. Their study is valuable, because it highlighted the critical issue of late HIV diagnosis and its impact on individuals.
Here just one question:
1) As noted by the author in line 48-50, a significant proportion of HIV-infected individuals sought medical help very late. A large percentage have already developed AIDS. This issue requires further discussion in this paper.
Reviewer 3 Report
Comments and Suggestions for Authors
This study mainly investigated the missed opportunities and their clinical repercussions in adults with an HIV new diagnosis or that were drug naïve. This is a retrospective, longitudinal, single-center cohort study. Some minor issues should be addressed.
1. Full names should be presented when abbreviations appeared for the first time. For example, HIV in Abstract.
2. Can authors provide the clearance number of ethical approval?
3. Proofreading is suggested. Some typos are found in the manuscript.
